# Beyond Utility: Evaluating LLM as Recommender

## ABSTRACT

With the rapid development of Large Language Models (LLMs), recent studies employed LLMs as recommenders to provide personalized information services for distinct users. Despite efforts to improve the accuracy of LLM-based recommendation models, relatively little attention is paid to beyond-utility dimensions. Moreover, there are unique evaluation aspects of LLM-based recommendation models, which have been largely ignored. To bridge this gap, we explore four new evaluation dimensions and propose a multidimensional evaluation framework. The new evaluation dimensions include: 1) history length sensitivity, 2) candidate position bias, 3) generation-involved performance, and 4) hallucinations. All four dimensions have the potential to impact performance, but are largely unnecessary for consideration in traditional systems. Using this multidimensional evaluation framework, along with traditional aspects, we evaluate the performance of seven LLM-based recommenders, with three prompting strategies, comparing them with six traditional models on both ranking and re-ranking tasks on four datasets. We find that LLMs excel at handling tasks with prior knowledge and shorter input histories in the ranking setting, and perform better in the re-ranking setting, beating traditional models across multiple dimensions. However, LLMs exhibit substantial candidate position bias issues, and some models hallucinate nonexistent items much more often than others. We intend our evaluation framework and observations to benefit future research on the use of LLMs as recommenders. The code and data are available at https://anonymous.4open.science/r/EvaLLMasRecommender-3118/.

## CCS CONCEPTS

• **Information systems → Personalization**.

## KEYWORDS

Large Language Model, Recommendation System, Multidimensional Evaluation.

## 1 INTRODUCTION

The applications of recommendation systems (RSs) are becoming increasingly widespread. Meanwhile, the emergence of Large Language Models (LLMs) and their outstanding performance on various NLP tasks [6, 7, 27, 63] have garnered great attention, creating a growing interest in applying LLMs to RSs as well. LLMs can be implemented in various stages of the RS pipeline, e.g., feature engineering [36, 57] and feature encoding [41, 51]. In this paper, we focus on the application of LLMs directly as recommenders. This approach introduces more significant changes to the traditional recommendation paradigm and thus may have more unknown impacts.

Previous work has explored the performance of LLMs as recommenders along multiple conventional evaluation dimensions. Palma et al. [39] conduct a detailed comparison of the performance of ChatGPT with that of various traditional models, focusing on recommendation accuracy, diversity, popularity bias, and novelty.

FairLLM [61], CFaiRLLM [11] and FairEvalLLM [9] concentrate on user-side fairness, while IFairLRS [23] and [10] cast a light on item-side fairness. Although this previous work covers many conventional dimensions, they are under different settings and no single effort has comprehensively assessed all of these aspects.

Moreover, we hold that these conventional concerns cannot fully reflect the performance of LLMs as recommenders because many novel characteristics introduced by LLMs are not considered by these conventional dimensions. Recommendations by LLMs differ from those by traditional models. LLMs have strong zero-shot [52], textual and generative [7, 63] capabilities, whereas traditional models are highly data-dependent and generally lack robust text-processing abilities. Therefore, LLM recommenders may vary in terms of generalization abilities and show different performance when textual information can be more efficiently incorporated. Additionally, LLMs exhibit certain issues that traditional models do not, such as position bias [54] and hallucinations [59, 66], which introduce new challenges.

In this paper, we propose a multidimensional evaluation framework, including two conventional dimensions, utility and novelty, and four our proposed LLM-related dimensions. We call attention to the four additional evaluation dimensions: 1) history length sensitivity: delving deeper into the generalization capabilities, 2) candidate position bias: quantifying the issue of position bias, 3) generation-involved performance: evaluating the textual and generative capabilities, 4) hallucination: focusing on the hallucination problem of LLMs. Among them, dimensions 1 and 2 can also be evaluated for traditional models, but these issues are less relevant for them. Dimensions 3 and 4 are unique to LLMs as recommenders. By introducing these LLM-centric evaluation dimensions, we can gain a more comprehensive understanding of LLM recommendations and their differences from traditional recommendation models.

With this framework, we conduct a multidimensional evaluation of seven LLMs, with three prompting strategies, exposing areas where LLMs excel and where they do not. In the ranking setting, LLMs demonstrate better novelty and excel in the domains where they possess more extensive knowledge and the cold-start scenario in terms of accuracy. LLM-generated profiles can capture key patterns of the user history. However, candidate position bias is significant, damaging recommendation quality. Hallucinations occur, posing threats to the user experience. In the re-ranking setting, LLMs show more outstanding performance than traditional recommenders in nearly all conventional dimensions with any history length. Though candidate position bias can still harm performance, the problem is partially alleviated compared to the ranking setting.

Our main contributions are as follows:

- We define and explore four evaluation dimensions beyond utility and novelty thoroughly based on the strengths and weaknesses of LLMs to observe how the characteristics of LLMs can impact recommendation performance.
- We propose a reproducible, multidimensional evaluation framework for LLM-based recommenders, covering both

ranking and re-ranking tasks. With this framework, we evaluate seven LLMs using three prompting strategies, including in-context learning method, and compare them against six traditional models across four datasets.

- In the four LLM-related dimensions, we gain seven interesting observations, providing a better understanding of LLMs as recommenders for future researches.

## 2 RELATED WORK

### 2.1 LLM as Recommender

The remarkable capabilities demonstrated by LLMs have sparked the interest of researchers in utilizing them for recommendation [26]. Early work established the now widely adopted LLM-as-recommender paradigm [34, 35] by converting recommendation data into natural language inputs and obtaining recommendation lists in natural language form from LLMs. Under this paradigm, prompt design is one of the crucial factors affecting performance. Some studies manually design various efficient prompt strategies [8, 46, 52] and in-context learning methods [20, 34, 53] through experimentation. Others propose automated prompt engineering processes [12, 47, 56] leveraging the reflective capabilities of LLMs, reinforcement learning methods, and so on. This work has already demonstrated promising performance in some scenarios, but due to the lack of training on recommendation data, the results theoretically still have the potential for additional enhancement.

To further improve the recommendation capabilities of LLMs, researchers try to tune LLMs on recommendation data. The main objective is to align the natural language space with the recommendation space and the item space [4, 5, 16, 62]. Work such as OpenP5 [15, 58] and POD [28] propose strategies for unifying various recommendation tasks using templates and training them jointly with the language model's loss. Another set of studies [21, 33, 64, 65] explores different item indexing methods, aiming to incorporate more collaborative information through indexing.

The above mentioned work primarily focuses on improvements in accuracy. Nevertheless, there are many other aspects that are equally noteworthy in recommendation systems. Therefore, a more comprehensive evaluation of LLMs as recommenders is required.

### 2.2 Evaluation of Recommender Systems

Accuracy is one important dimension in evaluating RSs, but it is not the only important dimension. In traditional recommendation scenarios, substantial work has been done to assess factors beyond accuracy that are also essential to user satisfaction and platform sustainability, which can be categorized into following dimensions: novelty [13, 24, 68], popularity bias [1, 2, 69, 70], diversity [22, 24, 32], and fairness [14, 31, 55].

Existing evaluation work on LLMs as recommenders often considers only these conventional evaluation dimensions. Palma et al. [39] investigate the performance of ChatGPT with various prompts in terms of novelty, popularity bias, and diversity in top-K recommendation, cold-start scenarios, and re-ranking tasks. Other work explores the fairness of LLMs as recommenders [30, 45]. FaiR-LLM [61] and CFaiRLLM [11] evaluate fairness among user groups

with different sensitive attributes, such as gender and age. FairEval-LLM [9] further extends the research by addressing intrinsic fairness. IFairLRS [23] focuses on item-side fairness, observing the fairness of items belonging to different categories, while Deldjoo [10] considers both individual and group-level item-side fairness.

However, evaluating only these conventional dimensions is insufficient to fully uncover the characteristics of LLMs as recommenders, as their introduction brings new opportunities and challenges. Researchers have assessed LLMs from various angles since their emergence [18, 29, 40, 49, 67], revealing multiple distinctive advantages and drawbacks of LLMs. These unique characteristics will cause LLM recommendations to exhibit distinct features compared to traditional ones. More specifically, one systematic bias that LLMs may exhibit is a particular preference for responses at certain positions [54], which might lead to an input position bias in candidate lists in recommendations [20]. The potential for hallucination [59, 66] may result in LLMs composing and recommending nonexistent items. Moreover, the text and generation capabilities of LLMs make it possible for them to create textual user profiles, which can also be integrated into the recommendation process and have an impact on the results. None of these aspects are a concern for traditional recommendation models; however, when using LLMs as recommenders, they need to be examined more thoroughly.

## 3 MULTIDIMENSIONAL EVALUATION

### 3.1 Framework Overview

We propose a multidimensional evaluation framework of LLM-as-recommender that encompasses two traditional dimensions and four newly introduced dimensions related to LLMs' characteristics. The overview of our framework is shown in Fig.1. Our framework adopts the commonly-used LLM-as-recommender paradigm and considers two specific tasks, ranking and re-ranking, the primary difference of which lies in the formation methods of the candidate sets. The task settings are described in detail in Section 3.2. Considering the cost, our framework supports evaluation on small sample datasets, with the sampling method of the datasets detailed in Section 3.3. After obtaining the recommendations from the LLM, our evaluation includes six dimensions: two conventional dimensions, utility and novelty, and four newly proposed dimensions. These four new dimensions focus on the potential new impacts that LLMs might bring to recommendations. Detailed descriptions of the evaluation dimensions are provided in Section 3.4.

### 3.2 Task Settings

**Table 1: Notations used in the task description.**

| | | | |
|---|---|---|---|
| $h_u$ | user history | $\mathcal{T}$ | item title |
| $C_{u,y}$ | candidate item set | $\mathcal{P}$ | prompt strategy |
| $y$ | next interacted item | $R_u$ | output recommended list |

Using LLMs as recommenders typically involves three steps: converting recommendation task and data into prompts, obtaining LLM inferred outputs, and extracting recommendation lists from those outputs. For input instructions, a common paradigm is to provide either off-the shelf or fine-tuned LLMs with the task instruction, user history $h_u$ and candidate item set $C_{u,y}$. The prompting strategies $\mathcal{P}$

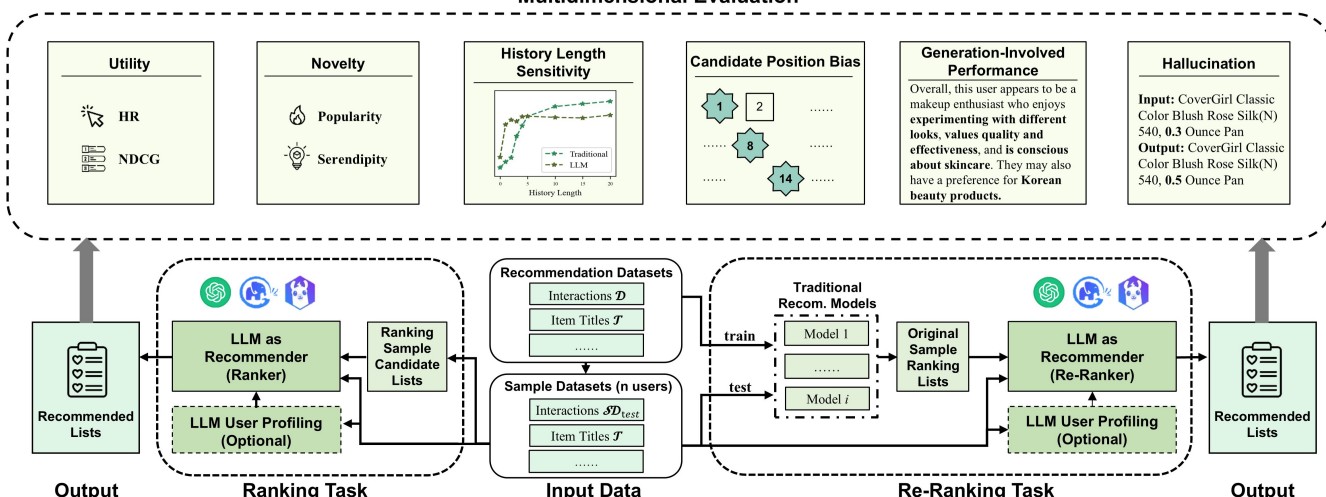

Figure 1: Multidimensional evaluation process of LLM as recommender.

usually index items by their titles $\mathcal{T}$. With the instructions, LLMs select top-K items $R_u$ from the candidate set for recommendations.

$$R_u^{g,\mathcal{P}} = g_{\text{LLM}}(\mathcal{P}(u, h_u, C_{u,y}; \mathcal{T})), (u, h_u, y) \in \mathcal{D}_{test} \quad (1)$$

More specifically, we support evaluations on two tasks: ranking and re-ranking, which are the two main settings in recommendation scenarios. We construct these tasks through different formation methods of the candidate sets.

**Ranking Task.** The ranking task aims to identify the top-K recommendations from the entire item set $\mathcal{I}$. However, limited by the input length of LLMs, we can only simulate this task using the candidate set. In this case, we sample m negative items randomly and combine them with the positive item to form $C_{u,y}^{rank}$.

$$C_{u,y}^{rank} = \{y, i_{x_1}, ..., i_{x_m}\}, i_{x_i} \in \mathcal{I}/h_u, i_{x_i} \neq y \quad (2)$$

**Re-Ranking Task.** The re-ranking task is the process of further refining and personalizing recommendations after obtaining preliminary results from some recommendation models. These models select the top-K items from $\mathcal{I}$, while re-ranking models reorder the items they have selected. In this case, $C_{u,y}^{rerank}$ should be the aggregation of the top-K recommendation of multiple traditional models. These recall models $f_i$ are trained on the training set $\mathcal{D}_{train}$, and required to make recommendations on $\mathcal{I}$ for each sample in $\mathcal{D}_{test}$.

$$C_{u,y}^{rerank} = R_u^{f_1} \cup R_u^{f_2} ... \cup R_u^{f_l} \quad (3)$$

### 3.3 Input Data

Any recommendation datasets that include the titles of items $\mathcal{T}$ and user-item interactions $\mathcal{D}$ can be used to conduct the evaluation for LLMs as recommenders in our framework.

Also, our framework supports evaluations on a small sample test set, considering that running inference on the full dataset can be time-consuming and costly in the context of LLMs. We utilize the leave-one-out strategy to divide $\mathcal{D}$ into $\mathcal{D}_{train}$, $\mathcal{D}_{valid}$ and $\mathcal{D}_{test}$, and randomly sample $n$ users from the entire user set $\mathcal{U}$ to form the sample test set $\mathcal{SD}_{test}$. When using the small sample setting, we

replace all $\mathcal{D}_{test}$ mentioned above with $\mathcal{SD}_{test}$, while the training set keeps unchanged as the entire $\mathcal{D}_{train}$.

To minimize the distributional difference between the sample and the full dataset, we perform a Kolmogorov-Smirnov (K-S) test on the hypothesis in (Eq. 4). Only samples with no significant differences will be passed on to further evaluation; otherwise, a new random sample is taken.

$$\begin{aligned} H_0 : &Metric(f_\Theta(\mathcal{D}_{test}; \mathcal{T}), \{y|(u, h_u, y) \in \mathcal{D}_{test}\}) \sim \\ &Metric(f_\Theta(\mathcal{SD}_{test}; \mathbf{T}), \{y|(u, h_u, y) \in \mathcal{SD}_{test}\}) \end{aligned} \quad (4)$$

where $f_\Theta$ represents a chosen traditional recommendation model, and $Metric$ denotes any utility or beyond-utility metrics.

### 3.4 Evaluation Dimensions

The evaluation covers six dimensions. Utility and novelty aspects are important areas in traditional RSs. We also conduct a more comprehensive evaluation of other beyond-utility aspects, including fairness and diversity. Due to coverage in previous work and limited space in this paper, we include them in the appendix. More notably, we take four additional dimensions into consideration, focusing on how the strengths and weaknesses of LLMs impact recommendations.

In the following part, we provide the detailed introduction of dimensions measured and metrics used. Considering that our evaluation should adapt to the scene of small samples, we carefully select and design metrics that can work on both large and small samples.

*3.4.1* ***Utility.*** Utility primarily represents the accuracy of recommendations. We utilize two widely-used metrics - *HR* and *NDCG*. *HR* measures the proportion of users obtaining accurate recommendations, while *NDCG* also takes the ranking quality of the recommendation results into consideration.

*3.4.2* ***Novelty.*** Beyond utility, we conduct a brief analysis of the novelty of the recommendations, to measure the abilities of models to recommend items that are not yet popular and accurately identify those that users might be interested in among them.

*Popularity: Average Percentage of Long Tail Items (APLT)* [1] measures the exposure probability of niche items.

$$APLT@K = \frac{1}{|\mathcal{R}|} \sum_{R_u \in \mathcal{R}} \frac{|R_u \cap \Phi|}{K} \quad (5)$$

where $\Phi$ is the set of long-tail items, i.e., items that are in the bottom 80% of $\mathcal{I}$ sorted by the item frequency. This bar is chosen according to the well-known Pareto Principle. A higher APLT indicates a greater chance that niche items will be recommended.

*Serendipity* [13] takes both unexpectedness and usefulness into consideration, which explicitly counts the amount of the correct recommendations that have been made by the model but not by the Most Popular (MostPop) method.

$$Serendipity@K = \frac{1}{|\mathcal{R}|} \sum_{R_u \in \mathcal{R}} \frac{1}{K} \sum_{i \in R_u} \mathbb{1}(i \notin R_u^{pop}) \quad (6)$$

The higher the value of the metric, the greater the unpredictability.

### 3.4.3 History Length Sensitivity.
In practice, there is often the issue of insufficient data, i.e., the cold start problem. However, it is expected that LLMs, with their world knowledge and generalization capability, should also perform relatively well when dealing with cold users. To gain deeper insights into this expectation, we introduce the evaluation dimension of history length sensitivity. We observe the recommendation accuracy with user history of different lengths as input. For testing, $h_u$ is truncated to a specified length $\mathcal{L}$, retaining the most recent $\mathcal{L}$ records. Additionally, at each history length, we also reduce the training set to $\mathcal{D}'$ as in (7), to ensure that off-the-shelf LLMs and trained models are exposed to the same amount of information.

$$\mathcal{D}' = \{(u, h_u', y) | (u, h_u, y) \in \mathcal{D}/\mathcal{D}_{test}, y \in h_{u,end(u)}[-\mathcal{L}:],$$
$$h_u' = h_u \cap h_{u,end(u)}[-\mathcal{L}:]\} \quad (7)$$

where $end(u)$ refers to the last interacted item of $u$, and $h_{u,end(u)}$ is the interaction history before the last interaction.

### 3.4.4 Candidate Position Bias.
Unlike traditional models, an inherent position bias of LLMs causes them to be very sensitive to the position of items in the candidate set when making recommendations. To illustrate, LLMs tend to place more priority on the candidates in the front of input candidate lists even if the prompt indicates that the candidates are sorted randomly. This preference for early positions affects the recommendation quality, since we cannot know the positive items in advance and place them at the front. Therefore, we design a metric to quantify this candidate position bias. The metric compares the recommendation accuracy when the positive items are randomly placed in the input candidate list to when they are always placed in the first position. Considering that both accuracy metrics (*HR* and *NDCG*) have an upper limit of 1, the range of variation for higher values is smaller than for lower values. To address this problem, we apply an inverse logarithmic transformation before calculating the difference in variation, allowing for a fairer comparison between higher and lower accuracy. Ideally, this metric should be zero.

$$CandDif_{Acc} = -log(1 - Acc(\mathcal{R}^{first\ position}))$$
$$-(-log(1 - Acc(\mathcal{R}^{random\ position}))), \ Acc = \{HR, NDCG\} \quad (8)$$

### 3.4.5 Generation-Involved Performance.
In this section, we mainly focus on the impact that introducing the generation of user profiles has on the performance of recommendations. Traditional user profiles are mostly existing information in the dataset or are represented as uninterpretable vectors. LLMs, with their strong textual and generative capabilities, can generate readable textual profiles based on user history. In this case, adding the LLM profiling step has the potential to enhance the explainability of recommendations and further improve the recommendations due to the extension of the logical chain [42]. Thus, we include the evaluation of the user profile generation-involved performance in our framework to delve deeper into how LLMs' capabilities can benefit RSs.

User profile generation is an optional step in our framework. If included, we first generate user profiles with a LLM based on the specified length of interaction history (Eq.9) and then incorporate them into prompts (or replace the original history) to let the LLM make recommendations. Afterward, we evaluate the differences between strategies with or without profile generation, as well as the differences among using profiles generated by different LLMs.

$$Profiles_u = g_{LLM}(\mathcal{P}_{profiling}(u, h_u[-\mathcal{L}:]; \mathcal{T})), (u, h_u, y) \in \mathcal{SD}_{test} \quad (9)$$

### 3.4.6 Hallucinations.
Hallucination in RSs refers to the phenomenon that LLMs may invent some items not present in $\mathcal{I}$ and recommend them. These items cannot be recommended to users in practice, doing so would compromise the results. For our evaluation, we implement a string matching algorithm ignoring case, space, and special symbols to parse the recommended lists $\mathcal{R}$ comprised of the titles of items outputted by LLMs. Items that fail to be matched are considered imaginary items. We propose that the proportion of items fabricated by LLMs can be used to measure the hallucination issue. The smaller the metric, the fewer the hallucinations.

$$Hallucination = \frac{1}{|\mathcal{R}|} \sum_{R_u \in \mathcal{R}} \frac{1}{K} \sum_{i \in R_u} \mathbb{1}(i \in \mathcal{I}) \quad (10)$$

## 4 EVALUATION RESULTS
## 4.1 Experimental Settings

### 4.1.1 Dataset.
We conduct extensive experiments on four real-world datasets, Amazon *All Beauty*, *Sports & Outdoors*, *Movielens-1M*, and *LastFM*. The user interaction history length in the first two datasets is shorter, while the length in the last two datasets is comparatively longer. In all datasets, item titles are used as the item descriptions to form the interaction history and candidate sets. We preprocess the four datasets with a 5-core filter and divide the datasets according to leave-one-out splitting. The detailed statistics of these datasets are presented in Tab.2.

**Table 2: Statistics of the experimental datasets.**

|  | #User | #Item | #Inter | #Density (%) |
|---|---|---|---|---|
| Beauty | 22363 | 12101 | 198502 | 0.0734 |
| Sports | 35598 | 18357 | 296337 | 0.0453 |
| ML-1M | 6040 | 3416 | 999611 | 4.8448 |
| LastFM | 1543 | 7286 | 173778 | 1.5458 |

*4.1.2 Models.* **LLM as Recommender.** We mainly evaluate the effectiveness of using off-the-shelf LLMs for recommendations in the zero or few-shot scenario. The evaluation includes seven LLMs that are capable of accomplishing this task of varying sizes, both open-source and closed-source:

- **GPT** [38] is an autoregressive language model that utilizes the Transformer architecture. In our evaluations, we adopt three versions, *gpt-3.5-turbo-0125*, *gpt-4o-mini-2024-07-18* & *gpt-4o-2024-08-06*.
- **Claude** [3] is characterized by its advanced capabilities in tasks requiring contextual comprehension and nuanced dialogue. We evaluate the *claude-3-haiku-20240307* version.
- **Llama** [48] is a well-known open-source LLM with a focus on scalability and performance. We experiment on *llama-3-70b-instruct*.
- **Qwen** [60] is excelling at handling long context lengths. We carry out experiments with *qwen-2-7b-instruct*.
- **Mistral** [37] utilizes the Grouped-Query Attention mechanism and is renowned due to its efficiency. We conduct experiments using *mistral-7b-instruct-v0.2*.

We implement three prompting strategies, including in-context learning.

- **OpenP5** [15, 58] applies a paradigm that unifies five recommendation tasks and is fine-tuned on a T5 backbone. We adopt their templates as the base version of our prompts.
- **LLMRank** [20] put forward two prompting strategies and one bootstrapping strategy. We choose the better-performing recency-focused strategy for performance comparison.
- **LLMSRec** [53] propose a method of aggregating multiple meticulously selected users into a demonstration example to further enhance LLMs through in-context learning.

Other off-the-shelf or fine-tuned LLMs capable of performing this task can also be evaluated by our framework.

**Traditional Models.** To further highlight the unique impacts of LLMs, we also compare the performance of LLMs as recommenders with that of traditional recommendation models. These traditional models are trained on the training set. Implemented traditional models can be divided into two categories: (1) traditional content-based recommendation models (BM25 [44]) and (2) traditional ID-based recommendation models (MostPop, BPRMF [43], GRU4Rec [19], SASRec [25] and LightGCN [17]). A brief introduction is given in the appendix.

*4.1.3 Implementation Details.* For LLMs as recommenders, we reference the templates of sequential recommendation task in OpenP5 [58] and add some output formatting texts to their templates to construct our base version of prompt templates. An example can be seen in Fig.7. Regarding the traditional models, our experiments are carried out using a popular library ReChorus [50]. For hyperparameters, we conduct a detailed parameter search to find the best config for each model (refer to the appendix). The results reported are the average of five experiments with different random initializations. We randomly sample 1000 test users under both settings. The hyperparameter settings can be found in the appendix.

**Ranking Setting.** Each user in the test set is provided with a candidate pool of 1 positive item and 19 randomly retrieved negative items. We will randomly shuffle the position of the 20 items in the prompts, but the order of which is kept the same among the experiments of different LLMs due to candidate position bias. The results are reported in Section 4.2-4.7.

**Re-Ranking Setting.** The size of a candidate pool is also set to 20 and the pools are formed based on the top-K items recommended by four models, BPRMF, GRU4Rec, SASRec and LightGCN. The order of each pool are kept unchanged as well. The results are reported in Section 4.8.

Due to page limitations, we only present a selection of representative results. The complete results can be found in the repo.

## 4.2 Utility

The utility of the recommendations from different LLMs and traditional models is reported in Tab.3. In Beauty, Sports, and ML-1M, LLMs demonstrate some recommendation capabilities; however, the overall recommendation accuracy of these models is inferior to that of traditional models. The two open-source models with 7b parameters cannot outperform the MostPop method, but the larger Llama3-70b and closed-source models achieve similar or better accuracy compared to MostPop. The best-performing LLM, GPT-4o, exhibits performance that even surpasses GRU4Rec in Sports, yet it still falls short of LightGCN.

More impressively, in LastFM, LLMs exhibit stronger recommendation capabilities. Several LLMs achieve significantly better recommendation accuracy than the best traditional model, Light-GCN. Since LLMs might have a better understanding of singers in the LastFM dataset due to the reason that these singers are more likely to have appeared in the training corpus, this result indicates that LLMs can effectively leverage their knowledge in areas where they excel and possess rich knowledge for recommendations. In this case, employing LLMs in recommendations can be beneficial to utilize their world knowledge.

In summary, for ranking tasks, LLMs have the potential to make better recommendations because of their world knowledge, while it is also crucial for LLMs to learn collaborative filtering information. The above results can be summarized as the first key observation:

*Observation 1. Overall, current LLMs are less accurate than traditional models, but they can exhibit greater accuracy in domains where they possess more extensive knowledge.*

## 4.3 Novelty

In this section, we evaluate whether niche items that users are interested in can be discovered. It might be assumed that when LLMs make recommendations, they tend to suggest popular items based on inherent stereotypes. However, our results show that off-the-shelf LLMs are far less affected by popularity bias than traditional models. In Fig.2, regarding *Popularity* performance, besides BM25, the other top three methods are all LLMs, while the strong performance of BM25 on this metric may be a tradeoff for its relatively lower accuracy. LLMs' understanding of popular trends does not cause them to focus on the biggest hit items in the dataset. In LastFM, GPT4o even demonstrates not only better accuracy but also less popularity bias at the same time. Moreover, this provision of exposure for niche items is not merely a fairness concern; LLMs can indeed accurately identify and recommend the correct niche items. Considering *Serendipity* in Fig.2, the metric that favors more accurate and unexpected results, although the accuracy of

**Table 3: Utility. The notation "P" refers LLM-generated user profile. "**" denotes the p<0.01 significance compared to the highest values of other groups.**

| | | Beauty | | Sports | | ML-1M | | LastFM | |
|---|---|---|---|---|---|---|---|---|---|
| | | HR@5 | NDCG@5 | HR@5 | NDCG@5 | HR@5 | NDCG@5 | HR@5 | NDCG@5 |
| Traditional | BM25 | 0.271 | 0.1734 | 0.274 | 0.1718 | 0.244 | 0.1341 | 0.276 | 0.1726 |
| | MostPop | 0.504 | 0.3434 | 0.498 | 0.3479 | 0.675 | 0.4765 | 0.650 | 0.5100 |
| | BPRMF | 0.652 | 0.5015 | 0.664 | 0.4958 | 0.843 | 0.6866 | 0.841 | 0.7645 |
| | GRU4Rec | 0.648 | 0.4846 | 0.651 | 0.4692 | 0.863 | 0.7150 | 0.827 | 0.7284 |
| | SASRec | 0.647 | 0.4996 | 0.686 | 0.5105 | **0.875**** | **0.7385**** | 0.836 | 0.7635 |
| | LightGCN | **0.673**** | **0.5177**** | **0.708**** | **0.5277**** | 0.853 | 0.6881 | **0.860** | **0.7667** |
| LLM | Mistral-7b | 0.246 | 0.1586 | 0.322 | 0.2029 | 0.327 | 0.2350 | 0.639 | 0.5104 |
| | Qwen2-7b | 0.353 | 0.2529 | 0.299 | 0.2128 | 0.346 | 0.2588 | 0.718 | 0.6681 |
| | Llama3-70b | 0.526 | 0.3882 | 0.557 | 0.4060 | 0.502 | 0.3680 | 0.846 | 0.7728 |
| | Claude3 | 0.370 | 0.2455 | 0.461 | 0.3131 | 0.620 | 0.4514 | 0.849 | 0.7694 |
| | GPT3.5 | 0.527 | 0.3799 | 0.593 | 0.4126 | 0.663 | 0.5010 | 0.866 | 0.7952 |
| | GPT4o-mini | 0.516 | 0.3757 | 0.575 | 0.4066 | 0.674 | 0.5083 | 0.873 | 0.7999 |
| | GPT4o | **0.604** | **0.4450** | **0.676** | **0.4922** | **0.734** | **0.5700** | **0.886** | **0.8159** |
| | GPT4o-mini P-only | 0.491 | 0.3337 | 0.575 | **0.4066** | **0.651** | **0.4796** | 0.765 | 0.6242 |
| | LLMRank (Llama3-70b) | 0.509 | 0.3954 | 0.514 | 0.3823 | 0.589 | 0.4493 | 0.880 | 0.8225 |
| | LLMSRec (Llama3-70b) | **0.553** | **0.4121** | **0.581** | 0.4057 | 0.528 | 0.3792 | **0.906**** | **0.8300**** |

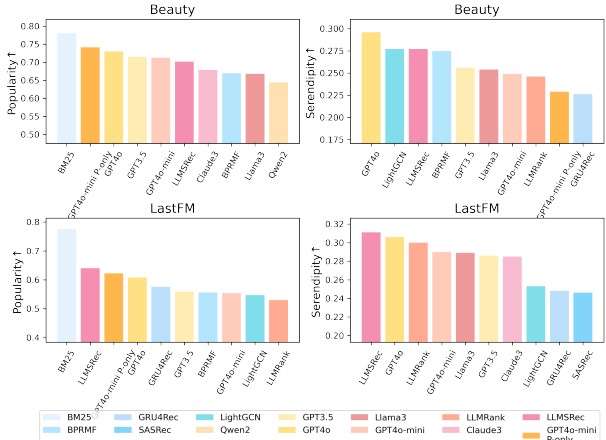

**Figure 2: Novelty Performances in Beauty and LastFM. The figures display the top 10 best-performing methods.**

LLMs' recommendations is inferior in Beauty, the *Serendipity* of the best-performing LLMs can still outperform traditional models. This result suggests a major advantage of LLMs, which is their ability to recommend less popular items, enhancing both fairness and accuracy.

*Observation 2. LLMs are adept at recommending more niche items correctly.*

## 4.4 History Length Sensitivity

The variation of recommendation accuracy with input history length is shown in Fig.3. Focusing on the performance of LLMs, we can conclude that they can effectively utilize user history data, as they perform much better with than without history (0 history length) as input, except for Mistral and Qwen2 in ML-1M. In Beauty and Sports, when user history is not provided, the HR@5 of LLMs is essentially equal to random recommendations (0.25); however, in ML-1M and LastFM, LLMs achieve a significantly higher HR@5 than 0.25 when the history length is 0, indicating that they already possess knowledge of movies and singers.

Nevertheless, the performance of the LLM does not consistently improve with the increasing length of input user history. The latest and second latest history inputs generally bring substantial improvements to the performance of LLMs, but the growth rate quickly slows down thereafter, and they often reach their peak performance with fewer than 10 history inputs. In comparison, traditional models require a longer history to achieve their best performance, keeping improving smoothly as the length of history increases. What is noteworthy is that, when there are only 1-2 historical records, best LLMs can demonstrate greater accuracy than traditional models. This suggests LLMs may offer insights into solving the cold start problem. However, it is also necessary to consider how to address the issue of LLMs not being able to utilize longer histories.

Another notable issue is that Qwen2 and Mistral exhibited abnormal performance in ML-1M. With user history input, their performance deteriorates. This may indicate that these 7b models experience a conflict between injected knowledge and their existing knowledge, leading to poorer results.

*Observation 3. LLMs require only brief histories to perform well, while longer histories do not always benefit LLMs. LLMs can beat the traditional models in the cold-start scenario.*

## 4.5 Candidate Position Bias

In this section, we observe the differences in the accuracy of users with positive items placed at different positions within the input candidate list. In general, candidate position bias is prevalent in LLM-based recommendations. According to Fig. 4, although the performance of traditional models and LLMs both fluctuates across different user groups when the positive item appears at different positions, traditional models remain relatively steady overall. In contrast, the performance of LLMs deteriorates as the positive item is placed further down the input list. This trend is particularly pronounced in Beauty, Sports, and ML-1M, and also observable in the LastFM dataset among Qwen2, Llama3, Claude3 and GPT4o-mini. The only exception is Mistral, which maintains relatively stable performance across the first three datasets; however, in LastFM, it exhibits a bias favoring items placed in later positions.

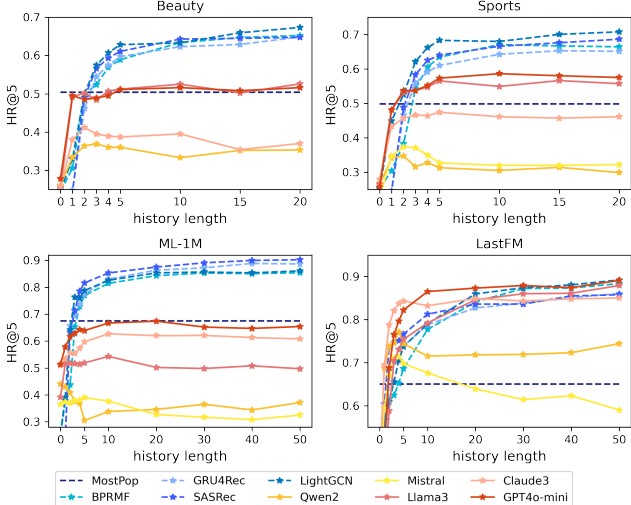

**Figure 3: History Length Sensitivity of the best traditional models (blue dashed line) and LLM-powered methods (yellow & red solid line).**

This candidate position bias can adversely affect the accuracy of recommendations, since it suggests that LLMs tend to make indiscriminate recommendations for the top-positioned item. When the top item is not positive, this indiscriminacy will affect the priority of the real positive items in the recommended lists. Therefore, mitigating this candidate position bias and fostering LLMs to rank items based on their actual relevance and quality are necessary to enhance accuracy. It is worth adding that previous work [20] has discovered that placing the positive items in the middle results in the highest accuracy. However, after we expand the user sample size and the number of datasets, our results show that most LLMs tend to prefer items positioned earlier in the list.

*Observation 4. LLMs suffer from a severe candidate position bias, favoring items at the beginning most of the time.*

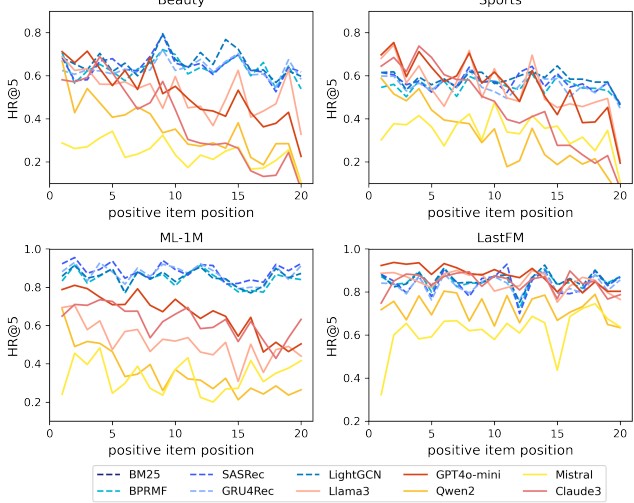

**Figure 4: Candidate Position Bias of LLM-based methods: recommendation accuracy when positive items placed at different positions within the candidate list.**

## 4.6 Generation-Involved Performance

In this part, we examine the multi-faceted performance when incorporating the LLM user profile generation step. For more details on the step, please refer to Section 3.4.5. When given the task of inferring a user's personality and preferences, LLMs can generate personalized user profiles based on the user's history. Fig.5 shows a summary of the user profile provided by GPT4o-mini of a user in Beauty. The LLM can identify the subcategories of products that the user prefers and the key factors they value, which can help enhance the understandability of the recommendation process.

In Beauty and Sports, the generated user profile contains most of the patterns useful for LLMs to make recommendations. According to Fig.6, we can see that removing the history and using only the generated profile, LLMs can achieve almost similar performance in terms of accuracy and better performance regarding popularity. Utilizing both the profile and history simultaneously leads to improvements across all four dimensions, regardless of history length. The trend in Sports is consistent with Beauty. In ML-1M and LastFM, LLMs can also generate personalized profiles that capture key patterns, but likely due to LLMs having deeper knowledge of specific items in ML-1M and LastFM, using the profile results in some loss of accuracy compared to using only the history.

*Observation 5. LLMs can generate user profiles that capture a majority of the key patterns useful for recommendations, which can help enhance the recommendation interpretability.*

> **User Profile:** Beauty user 6 is a cosmetics enthusiast with a keen interest in makeup products, skincare, and hair care. They display a preference for a diverse range of products, showing a willingness to experiment with different brands and styles. The user's focus on skincare suggests an awareness of health and aesthetics, while their selection of various palettes reveals a love for creativity in makeup. Overall, they demonstrate an appreciation for both performance and value in their beauty purchases.

**Figure 5: An example of LLM-Generated Profile.**

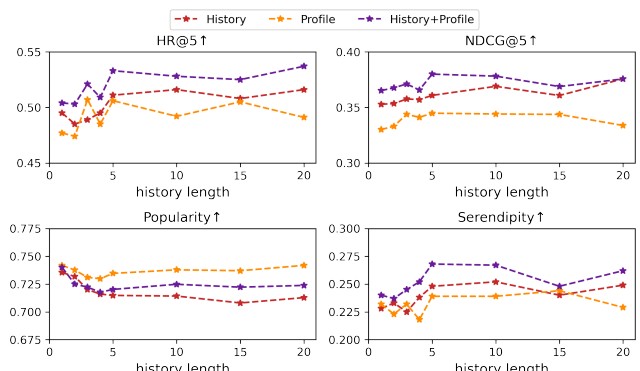

**Figure 6: Utilizing LLM-generated user profiles with different history lengths in Beauty with GPT4o-mini.**

## 4.7 Hallucinations

This section examines the issue of hallucinations in LLM-based recommendations. When dealing with ranking tasks, LLMs still exhibit an overall deficiency in following instructions. In Tab.4, we can see that the proportion of non-existent items generated

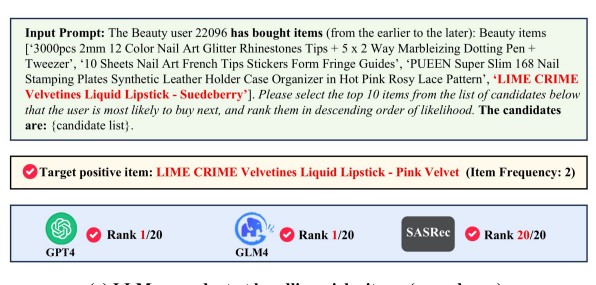

**Input Prompt:** The Beauty user 22096 **has bought items** (from the earlier to the later): Beauty items ['3000pcs 2mm 12 Color Nail Glitter Rhinestones Tips + 5 x 2 Way Marbleizing Dotting Pen + Tweezer', '10 Sheets Nail Art French Tips Stickers Form Fringe Guides', 'PUEEN Super Slim 168 Nail Stamping Plates Synthetic Leather Holder Case Organizer in Hot Pink Rosy Lace Pattern', **'LIME CRIME Velvetines Liquid Lipstick - Suedeberry']**. *Please select the top 10 items from the list of candidates below that the user is most likely to buy next, and rank them in descending order of likelihood.* **The candidates are:** {candidate list}.

✅ Target positive item: **LIME CRIME Velvetines Liquid Lipstick - Pink Velvet** (Item Frequency: 2)

GPT4 ✅ Rank 1/20    GLM4 ✅ Rank 1/20    SASRec ✅ Rank 20/20

(a) LLMs are adept at handling niche items (a good case).

**Input Prompt:** The Beauty user **has bought items** (from the earlier to the later): Beauty items ['Godefroy Double Lash and Brow Treatment, for longer & thicker eyelash and eyebrows (3ml + applicator)', 'CND: Shellac Nourishing Remover, 32 oz', 'Coanir Facial Sauna System with Timer', 'Cricket Hair Brush Static Free, Mini Fast Flo, 1.28 Ounce']. *Please select the top 10 items from the list of candidates below that the user is most likely to buy next, and rank them in descending order of likelihood.* **The candidates are:** {candidate list}

✅ Target positive item: **Nail Soakers - 10pcs** (Item Frequency: 64)
❌ Confusing negative candidate item: **Organic Eye Gel - Best Eye Firming Treatment - Dark Circle, Wrinkle and Puffiness Treatment - Best Eye Cream - Anti Aging - Soothes and Restores Skin Elasticity - Rejuvenates the Skin to Be Firmer and Younger Looking - Soul Vista Organic Eye Gel with Firming Peptides 0.5 Oz (15 Ml)** (Item Frequency: 7)

GPT4 ✅ Rank 10+/20    ❌ Rank 1/20
GLM4 ✅ Rank 10/20    ❌ Rank 1/20
SASRec ✅ Rank 4/20    ❌ Rank 18/20

(b) LLMs are influenced by clickbait titles (a bad case).

**Figure 7: Case study on the strengths and the weaknesses of LLMs as recommenders.**

is mostly around 1% and less than 5% in models other than Mistral, while Mistral outputs quite a lot of non-existent items, with a proportion around 23% in Beauty. Common errors made by LLMs may include omitting part of the original titles, combining multiple titles, or misstating elements such as capacity. When the first two types of errors occur, it is difficult to map the generated item back to the corresponding item in the original dataset. A well-designed mapping strategy is highly needed; otherwise, it will result in many invalid recommendations. Hallucination will be one of the critical factors affecting the recommendation quality of Mistral. Although the hallucination issue is relatively less severe in other models, around 1% occurrence can still impact the user experience.

***Observation 6. In general, most LLMs tend to generate below 5% of non-existent items, while some LLMs significantly hallucinate more items.***

**Table 4: Hallucinations of different LLMs.**

|        | Mistral | Qwen2  | Llama3 | Claude3 | GPT4o-mini | GPT4o  |
|--------|---------|--------|--------|---------|------------|--------|
| Beauty | 0.2288  | 0.0166 | 0.0344 | 0.0088  | 0.0096     | 0.0046 |
| Sports | 0.1204  | 0.0228 | 0.0456 | 0.0062  | 0.0100     | 0.0042 |
| ML-1M  | 0.2020  | 0.0348 | 0.0216 | 0.0658  | 0.0032     | 0.0036 |
| LastFM | 0.0412  | 0.0074 | 0.0014 | 0.0026  | 0.0018     | 0.0026 |

## 4.8 LLM as Re-Ranker

**Table 5: Re-rank performance in Beauty. The bold numbers indicate the best performance. "\*\*" represents the best show a significant difference to the second at a 0.01 p-value level.**

|                          | BPRMF  | GRU4Rec | SASRec | LightGCN | GPT3.5     |
|--------------------------|--------|---------|--------|----------|------------|
| $HR@5\uparrow$           | 0.0238 | 0.0216  | 0.0242 | 0.0244   | **0.0320**\*\* |
| $NDCG@5\uparrow$         | 0.0145 | 0.0130  | 0.0144 | 0.0147   | **0.0202**\*\* |
| $Popularity\uparrow$     | 0.0219 | 0.0086  | 0.0259 | 0.0226   | **0.0904**\*\* |
| $Serendipity\uparrow$    | 0.0140 | 0.0106  | 0.0144 | 0.0132   | **0.0200**\*\* |
| $Cand.\ Pos.\ Bias_{HR}\downarrow$   | **0.0000** | **0.0000** | **0.0000** | **0.0000** | 0.2282\*\* |
| $Cand.\ Pos.\ Bias_{NDCG}\downarrow$ | **0.0000** | **0.0000** | **0.0000** | **0.0000** | 0.2856\*\* |
| $Hallucination\downarrow$ | -      | -       | -      | -        | 0.0159     |

Next, we assess LLMs' performance on the re-ranking task from the above multiple dimensions. According to Tab.5, GPT-3.5 already shows higher re-ranking performance than four traditional models, both enhancing accuracy and promoting the exposure of niche items. One reason might be that LLMs both explicitly and implicitly utilize new features beyond those used by the four traditional models to rank the candidates, thereby better distinguishing between items in the candidate set, while the four traditional models find it more difficult to re-rank their own top recalled items. In practice, it might be a viable option to apply LLMs in the re-ranking stage.

The candidate position bias issue is similarly alleviated. In this scenario, considering that some candidate sets do not contain positive items, we calculate $CandDif$ based only on samples that include positive examples. Compared to $CandDif_{HR}$ and $CandDif_{NDCG}$ of 0.5172 and 0.4713 in ranking setting in Beauty, GPT-3.5 also shows relative insensitivity to position when re-ranking (Tab.5). Moreover, we can actually design the order of candidates according to the original recall model's ranking to leverage this position bias for better recommendations in the re-ranking task.

**Observation 7. Compared to ranking tasks, LLMs are better at re-ranking tasks regarding utility and beyond.**

## 5 CASE STUDY

We select two examples to provide a more intuitive illustration of the strengths and weaknesses of LLMs as recommenders. Fig.7(a) shows a case that LLMs are good at. LLMs can utilize the similarity of the titles between the interacted items and the target item to pick out the right candidate successfully, while due to the low popularity of the target item, traditional models perform much worse. However, a typical drawback of LLMs as recommenders utilizing text information is demonstrated in Fig.7(b), which is "clickbait". LLMs apparently have little ability to resist the allure of a title that is long and complex. The multiple elements it contains, such as "natural ingredients", "large capacity", and "multi-functionality" tend to make LLMs allocate the top priority to it always.

## 6 CONCLUSIONS

Existing evaluations of LLMs as recommenders ignore LLM-specific aspects. We define and explore four new evaluation dimensions that reflect the unique characteristics of LLM-based recommendation more thoroughly, as compared to traditional approaches. These dimensions consider history length, candidate position bias, user profile generation, and hallucinations. We evaluate seven LLM-based recommendation systems and six traditional approaches under these new dimensions, along with the two traditional dimensions of utility and novelty. We explore both ranking and re-ranking settings. In the ranking setting, LLMs perform impressively in the domains they are familiar with and when the input history length is relatively short. However, LLMs suffer from severe candidate position bias. In the re-ranking setting, LLMs demonstrate better performance across multiple dimensions. The observations suggest intriguing future directions, such as leveraging longer histories, item mapping strategy, etc. We welcome interested researchers to use and improve our reproducible evaluation framework. Code and data are available at https://anonymous.4open.science/r/EvaLLMasRecommender-3118/.

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

# A APPENDICES

## A.1 Metric Definition

In the main body of the paper, we have introduced a selection of metrics. And here, we provide an panoramic view to all the evaluation metrics implemented in our framework. Tab.6 displays frequently used notations in the following parts. For more comprehensive results, refer to our github repo.

**Table 6: Frequently-used Notations**

| Notation | Explanation |
|---|---|
| $\mathcal{U}/\mathcal{SU}$ | user set / sample user set |
| $\mathcal{I}$ | item set |
| $Cand_u$ | candidate set of user $u$ |
| $\mathcal{R}_u$ | top-K recommendation of user $u$ |
| $Pop_i$ | global popularity of item $i$ |
| $Y(u, i)$ | 1 when user $u$ has interacted with item $i$, otherwise 0 |

**Utility Metrics.**

$$NDCG@K = \frac{1}{|\mathcal{SU}|} \sum_{u \in \mathcal{SU}} \sum_{i \in \mathcal{R}_u} \frac{Y(u, i)}{log_2(r_{u,i} + 1)} \quad (11)$$

where $r_{u,i}$ represents the rank of the item $i$ in $\mathcal{R}_u$.

**Novelty Metrics.** Novelty measures whether the recommended content would be difficult to be discovered by users without recommendation systems. Thus, good novelty can enhance user engagement. Two metrics, *Self-information*[68] is related to the average probability of each item in the recommended lists being known by the users, while *Serendipity*[13] takes both unexpectedness and usefulness into consideration.

$$SelfInformation@K = \frac{1}{|\mathcal{SU}|} \sum_{u \in \mathcal{SU}} \frac{1}{K} \sum_{i \in \mathcal{R}_u} log_2 \frac{|\mathcal{U}|}{|\mathcal{U}_i|} \quad (12)$$

where $\mathcal{U}_i$ refers to all the users that have interacted with item $i$. The self-information of an item is defined by the reciprocal of the chance of randomly selecting a user and the user has interacted with the item.

We quantifies *Serendipity* by comparing the output of recommendation models to the results of Most Popular method.

$$Serendipity@K = \frac{1}{|\mathcal{R}|} \sum_{R_u \in \mathcal{R}} \frac{1}{K} \sum_{i \in R_u} \mathbb{1}(i \notin R_u^{pop}) \quad (13)$$

Both metrics being higher indicates better unpredictability.

**Popularity Metrics.** To formulate the item popularity of the recommendations, we select three metrics focused on three distinctive perspectives. To be specific, *Average Recommendation Popularity (ARP)* shows the average item popularity of the top-k recommended lists, *Average Percentage of Long Tail Items (APLT)*[1] quantifies the percentage of long-tailed items appearing in the lists. And *Popularity-based Ranking-based Equal Opportunity (PopREO)*[70] measures the difference between true positive rate of items with different levels of popularity, which can reflect whether popular

items are more likely to be correctly recommended.

$$ARP@K = \frac{1}{|\mathcal{R}|} \sum_{R_u \in \mathcal{R}} \frac{1}{K} \sum_{i \in \mathcal{R}_u} Pop_i \quad (14)$$

$$APLT@K = \frac{1}{|\mathcal{R}|} \sum_{R_u \in \mathcal{R}} \frac{|R_u \cap \Phi|}{K} \quad (15)$$

where $\Phi$ in *APLT* is a subset of $\mathcal{I}$, items in which are the bottom 80% by the popularity. The bar is chosen according to the well-known Pareto Principle, also called 80/20 rules, that is a small portion, 20%, of the products usually earns a large portion of income. In recommendation scenario, the phenomenon is that the leading popular items are also pruned to be recommended and dominate over less popular items, which will lead to the unsatisfactory of users with niche hobbies. Therefore, suitable exposure of niche items is necessary.

$$PopREO@K = \frac{std(P(R@k|g = g_1, y = 1)...P(R@k|g = g_5, y = 1))}{mean(P(R@k|g = g_1, y = 1)...P(R@k|g = g_5, y = 1))} \quad (16)$$

$$where \ P(R@k|g = g_p, \ y = 1) = \frac{\sum_{u=1}^{|\mathcal{SU}|} \sum_{i=1}^{K} G_{g_p}(R_{u,i}) Y(u, \mathcal{R}_{u,i})}{\sum_{u=1}^{|\mathcal{SU}|} \sum_{i \in T_u} G_{g_p}(i) Y(u, i)} \quad (17)$$

where $R@K$ represents *Recall@K*, $T_u$ is the target positive items of user $u$, $G_{g_p}(i)$ is a function that returns 1 when $i \in g_p$. $P(R@k|g = g_p, \ y = 1)$ demonstrates the true positive rate of $g_p$, which should be the same among all the groups optimally. In this case, we equally divide the items into 5 groups according to their popularity to calculate *PopREO* and lower values of *PopREO* reflect a less biased recommendation regarding item popularity.

**Diversity Metrics.** As the phenomenon of information cocoons emerges and garners increased attention, diversity has also become an important evaluation aspects. A system capable of generating diverse results enables people to encounter a wider range of information beyond what they already know. In our evaluation, we utilize three diversity metrics. *Item Coverage* measures the diversity of items in a global view; *Overlap Item Coverage(OIC)* shows the inter-list diversity between the users.

$$ItemCoverage = \frac{|\mathcal{R}_1 \cup \mathcal{R}_2 \cup ... \cup \mathcal{R}_{|\mathcal{U}|}|}{|Cand_1 \cup Cand_2 \cup ... \cup Cand_{|\mathcal{U}|}|} \quad (18)$$

*Item Coverage* simply measures the proportion of candidate items that appear in the recommended lists. A higher metric indicates a better global diversity.

$$OIC@K = \frac{\sum_{u,v \in \{(u,v) | |Cand_u \cap Cand_v| > 0\}} \frac{|\mathcal{R}_u \cap \mathcal{R}_v|}{|Cand_u \cap Cand_v|}}{|\{(u,v) | |Cand_u \cap Cand_v| > 0\}|} \quad (19)$$

*Overlap Item Coverage (OIC)* measures the possibility of LLMs to recommend the same items to the users if there are items appear in the candidate lists of two users. High OIC suggests a tendency to recommend similar items to different users, indicating poor personalization and inter-list diversity.

**Fairness Metrics.** With regard to fairness, both user-side and item-side fairness should be paid careful attention. *Gini Coefficient*

is one of the most commonly used fairness metrics, which we implement to evaluate the individual item-side fairness. Since *PopREO* has already provided insights into the difference between groups of popular items and unpopular items, we omit the perspective of group-level item-side fairness in this part. When it comes to user-side fairness, we utilize *Demographic Parity Difference(DPD)*[49] to cast light on the group-level fairness between active users and inactive users and *Jain's Index* to quantify whether users obtain consistently high utility outcomes at the individual level.

$$\text{Gini@K} = \frac{\sum_{i_x, i_y \in I} |Pop_{i_x}^{\mathcal{R}} - Pop_{i_y}^{\mathcal{R}}|}{2|\mathcal{I}| \sum_{i \in I} Pop_i^R} \tag{20}$$

where $Pop_i^{\mathcal{R}}$ indicates the frequency of item $i$ in all the output recommended lists. It measures the area between the Lorenz curve (which represents the distribution of resource) and the line of perfect equality. The smaller the metric, the fairer.

$$DPD@K = |\mathbb{E}(NDCG@K|u \in U_A) - \mathbb{E}(NDCG@K|u \in U_I)| \tag{21}$$

where $U_A$ and $U_I$ are active user set and inactive user set divided by the median of the interaction history length of the users. A smaller metric represents a fairer treatment of the users.

$$Jain's\ Index = \frac{(\sum_{u \in \mathcal{SU}} NDCG_u@K)^2}{|\mathcal{SU}| \sum_{u \in \mathcal{SU}} NDCG_u@K^2} \tag{22}$$

Jain's Index primarily measures whether users have obtained consistently high utility outcomes. The closer to 1, the better the performance regarding user-side fairness.

**Candidate Position Bias Metrics.**

$$CandDif_{Acc} = -log(1 - Acc(\mathcal{R}^{first\ position}))$$
$$-(-log(1-Acc(\mathcal{R}^{random\ position}))),\ Acc = \{HR, NDCG\} \tag{23}$$

Both of the metrics should be zero ideally.

**Hallucination Metrics.**

$$Hallucination = \frac{1}{|\mathcal{R}|} \sum_{R_u \in \mathcal{R}} \frac{1}{K} \sum_{i \in R_u} \mathbb{1}(i \in \mathcal{I}) \tag{24}$$

The optimal value for this metric is 0.

## A.2 Introduction of Traditional Models

- **MostPop** recommends items according to their popularity.
- **BM25** [44], a ranking function built on TF-IDF. In this scenario, we treat the interaction history as documents and the candidate items as queries.
- **BPRMF** [43] is a model that optimizes a pairwise ranking loss function based on implicit feedback data.
- **GRU4Rec** [19] is a sequential recommending algorithm based on RNN structure dealing with short session data.
- **SASRec** [25] is a model that utilizes the self-attention architecture to capture sequential patterns in user behavior.
- **LightGCN** [17] learns the user and item embeddings through user-item interaction graph and includes only the most essential parts in GCN.

## A.3 Hyperparameter Settings

We performed a detailed search for the hyperparameters of each model. The hyperparameter settings for the reported version are as follows:

- **BM25.** In all four datasets, k1 and b are set to 1.5 and 0.75, respectively.
- **BPRMF.** In Beauty, we set the learning rate to $1e-3$, l2 to $5e-6$ and embedding size is assigned 64. In Sports, we set the learning rate to $1e-3$, l2 to $1e-6$ and embedding size is assigned 128. In ML-1M, we set the learning rate to $5e-4$, l2 to $1e-5$ and embedding size is assigned 64. In LastFM, we set the learning rate to $1e-3$, l2 to $5e-5$ and embedding size is assigned 128.
- **GRU4Rec.** In Beauty, we set the learning rate and l2 to $5e-3$ and $1e-4$, respectively. The size of hidden vectors in GRU and embedding vectors are 100 and 128. In Sports, we set the learning rate and l2 to $1e-3$ and $1e-4$, respectively. The size of hidden vectors in GRU and embedding vectors are 100 and 64. In ML-1M, we set the learning rate and l2 to $1e-3$ and $5e-5$, respectively. The size of hidden vectors in GRU and embedding vectors are 100 and 128. In LastFM, we set the learning rate and l2 to $1e-2$ and $1e-4$, respectively. The size of hidden vectors in GRU and embedding vectors are 100 and 128.
- **SASRec.** In Beauty, we set the learning rate and l2 to $1e-4$ and $5e-4$. The number of self-attention layers and the number of attention heads are configured as 1 and 1. The dimension of embedding vectors is 128. In Sports, we set the learning rate and l2 to $1e-4$ and $5e-4$. The number of self-attention layers and the number of attention heads are configured as 1 and 1. The dimension of embedding vectors is 128. In ML-1M, we set the learning rate and l2 to $1e-4$ and $1e-4$. The number of self-attention layers and the number of attention heads are configured as 2 and 1. The dimension of embedding vectors is 128. In LastFM, we set the learning rate and l2 to $1e-3$ and $5e-4$. The number of self-attention layers and the number of attention heads are configured as 1 and 1. The dimension of embedding vectors is 128.
- **LightGCN.** In Beauty, we set the learning rate and l2 to $1e-3$ and $1e-6$. We choose to use 3 layers and 128 as the dimension of embedding vectors. In Sports, we set the learning rate and l2 to $1e-3$ and $1e-6$. We choose to use 3 layers and 128 as the dimension of embedding vectors. In Beauty, we set the learning rate and l2 to $1e-3$ and $1e-6$. We choose to use 4 layers and 128 as the dimension of embedding vectors. In Beauty, we set the learning rate and l2 to $1e-3$ and $1e-5$. We choose to use 4 layers and 128 as the dimension of embedding vectors.

