# OpenReview forum: "Beyond Utility: Evaluating LLM as Recommender"
_ACM.org/TheWebConf/2025/Conference — WWW 2025 Poster_

### Official Review · Reviewer_Xto6 · 2024-11-26

**Novelty:** 3
**Technical Quality:** 4

**Review:**

This paper proposes a multidimensional evaluation framework for assessing Large Language Models (LLMs) as recommenders. The framework introduces four new dimensions: history length sensitivity, candidate position bias, generation-involved performance, and hallucinations, which are particularly relevant to LLM-based recommendation systems.

The paper has the following strengths:

+ The studied problem, new metrics for LLM-based RS, is very important.
+ The proposed metrics make sense and are indeed crucial to consider.
+ Very extensive experiments on various LLMs are used to derive interesting insights.

However, I think the main issue lies in the novelty of the introduced "new" dimensions. Most of the dimensions have been explored by previous work. For instance, candidate position bias has been discussed in studies like "Large Language Models are not stable recommender systems" and "Large language models are zero-shot rankers for recommender systems." Similarly, the hallucination aspect has been discussed in "Large Language Models for Generative Recommendation: A Survey and Visionary Discussions." While the paper brings these dimensions together in a unified metric bundle, it could further differentiate its contributions by exploring truly new dimensions or providing deeper insights into how to address these existing challenges.

**Questions:**

Please address the weakness in my main review.

**Reviewer Confidence:**

3: The reviewer is confident but not certain that the evaluation is correct

**Scope:**

4: The work is relevant to the Web and to the track, and is of broad interest to the community

---

### Official Review · Reviewer_39Zc · 2024-11-27

**Novelty:** 5
**Technical Quality:** 5

**Review:**

This paper proposes a multidimensional evaluation framework for LLM-based recommendation models, introducing four novel evaluation dimensions: history length sensitivity, candidate position bias, generation-involved performance, and hallucinations. The framework highlights unique challenges in LLM-based systems and evaluates seven LLM-based recommenders against six traditional models across ranking and re-ranking tasks on four datasets.

Pros:

1.The introduction of four LLM-specific evaluation dimensions is novel and addresses underexplored aspects of LLM-based recommendation systems.

2.Comprehensive experiments involving multiple datasets, prompting strategies, and comparison with traditional models add robustness to the findings.

3.The paper offers practical insights into LLM strengths, such as handling shorter input histories and excelling in re-ranking tasks.

Cons:

1.The comprehensiveness and independence of the metrics still need further exploration, such as explainability.etc.

2.The paper uses a lot of symbols, and listing them in a "Notations Used" table may harm readability.

**Questions:**

1.For the input data, does randomly sampling 𝑛 users from the entire user set U to form the test set SD𝑡𝑒𝑠𝑡 risk losing rare items and user diversity, given the typically long-tailed data distribution?

2.The meaning of "Generation-Involved Performance" is unclear, and it's not obvious why it should be used as a metric. Could the authors clarify this?

**Reviewer Confidence:**

3: The reviewer is confident but not certain that the evaluation is correct

**Scope:**

4: The work is relevant to the Web and to the track, and is of broad interest to the community

---

### Official Review · Reviewer_rDHt · 2024-12-02

**Novelty:** 5
**Technical Quality:** 5

**Review:**

This paper have introduced four novel evaluation dimensions and a multidimensional framework is a valuable addition to the research on LLM-based recommenders. It covers the ongoing issues with the LLMs such as position bias and hallucinations.

Strengths:

- Intrduction of four novel evaluation dimensions
- Proposes a clear and structured multidimensional evaluation framework, integrating both traditional (utility, novelty) and novel (position bias, hallucination) dimensions.
- Introduces innovative metrics tailored for LLM-specific challenges, such as hallucination detection and candidate position bias.


Weaknesses:

- There is no comparison or discussion if fine tuning of LLMs can solve the issues of bias and hallucinations.
- No discussion on how conventional RS models (e.g., collaborative filtering, matrix factorization) perform on the evaluation metrics

**Questions:**

Question 1: To what extent can fine-tuning LLMs mitigate inherent biases, such as candidate position bias, and reduce hallucination effects in recommendation tasks?
Question 2: Are the proposed evaluation dimensions uniquely suited to LLM-based recommendation systems, or could they be generalized to assess other advanced recommendation systems?
Question 3: Can the proposed evaluation framework and its dimensions be extended to non-recommendation tasks, such as classification or clustering, to assess similar performance aspects?

**Reviewer Confidence:**

3: The reviewer is confident but not certain that the evaluation is correct

**Scope:**

3: The work is somewhat relevant to the Web and to the track, and is of narrow interest to a sub-community

---

### Official Review · Reviewer_CKD3 · 2024-12-02

**Novelty:** 3
**Technical Quality:** 3

**Review:**

pros:
This paper proposed four evaluation matrices especially designed for the LLM based RS, test them on ranking and re-ranking tasks and got a lot of interesting observations.

Cons:
1) The newly proposed evaluation metrics are quite interesting. One of my concerts is Hallucinations. This is a general problem for LLM, not related to the recommendation task. Besides, before proposing the evaluation metrics, two things should be clarified:  is it a must to use LLM as RS model, only using LLM to generate something to assist the RS is not enough?
2) In general, people usually claim LLM works better than traditional RS model in cold-start settings. However, the results in Table 3 shows that LLM models get worse results on sparse datasets: Beauty and Sports. Could you further explain this?

**Questions:**

Please refer to Cons.

**Reviewer Confidence:**

3: The reviewer is confident but not certain that the evaluation is correct

**Scope:**

4: The work is relevant to the Web and to the track, and is of broad interest to the community